# Increased error-correction leads to both higher levels of variability and adaptation

**Elisabeth B. Knelange**, **Joan López-Moliner** *

Department of Cognition, Development and Psychology of Education, Vision and Control of Action (VISCA) Group, Institut de Neurociències, Universitat de Barcelona, Barcelona, Catalonia, Spain

* j.lopezmoliner@ub.edu

## Abstract

In order to intercept moving objects, we need to predict the spatiotemporal features of the motion of both the object and our hand. Our errors can result in updates of these predictions to benefit interceptions in the future (adaptation). Recent studies claim that task-relevant variability in baseline performance can help adapt to perturbations, because initial variability helps explore the spatial demands of the task. In this study, we examined whether this relationship is also found in interception (temporal domain) by looking at the link between the variability of hand-movement speed during baseline trials, and the adaptation to a temporal perturbation. 17 subjects performed an interception task on a graphic tablet with a stylus. A target moved from left to right or vice versa, with varying speed across trials. Participants were instructed to intercept this target with a straight forward movement of their hand. Their movements were represented by a cursor that was displayed on a screen above the tablet. To prevent online corrections we blocked the hand from view, and a part of the cursor's trajectory was occluded. After a baseline phase of 80 trials, a temporal delay of 100 ms was introduced to the cursor representing the hand (adaptation phase: 80 trials). This delay initially caused participants to miss the target, but they quickly accounted for these errors by adapting to most of the delay of the cursor. We found that variability in baseline movement velocity is a good predictor of temporal adaptation (defined as a combination of the rate of change and the asymptotic level of change after a perturbation), with higher variability during baseline being associated with better adaptation. However, cross-correlation results suggest that the increased variability is the result of increased error correction, rather than exploration.

## Introduction

The human race is remarkably skilled in timing tasks like catching or hitting a ball [1, 2]. However, large inter-individual differences can be observed when it comes to these skills. To become successful at ball sports, a person needs to be able to accurately account for continuous changes in both the environment and their own body. To this end, we can use sensory feedback to guide our movement [3]. However, sensory feedback from our movements is processed by our brain with a delay that can reach up to about 150 ms [4–6]. This can lead to

**Funding:** This work is part of the PACE project, which received funding from the European Union's Horizon 2020 research and innovation programme under the Marie Sklodowska-Curie grant agreement No 642961. We also received funding from grant PSI2017-83493-R of AEI/FEDER, UE. There were no commercial relationships. The funders had no role in study design, data collection and analysis, decision to publish, or preparation of the manuscript.

**Competing interests:** The authors have declared that no competing interests exist.

unstable movements, because corrections to errors are initiated with a delay. In order to not only rely on sensory feedback and select appropriate motor commands, we are thought to be able to predict the sensory consequences of these motor commands. Studies have shown that our brain maintains an internal forward model that holds the predictions regarding the sensory consequences [7, 8]. However, the outcome of a motor command can change due to systematic disturbances like fatigue in the muscles [9]. Accurate timing of our motor commands in a dynamic world requires continuous evaluation and updating of the predicted consequences of these motor commands [10]. This can be done by evaluating the difference between the actual and the predicted sensory consequences of a motor command (the prediction error [7, 11]). However, prediction errors do not always stem from incorrect motor commands. Noise can come from different stages of the sensorimotor process [12, 13]. Different types of noise associated with neural processes like movement planning [14–18] or execution [19–21] cause variability. In addition, trial-to-trial variability in the outcome of the movement can be caused by inaccurate estimates of the task requirements [22] and disturbances from the outside world [23]. Errors caused by these types of noise do not necessarily require updates of the internal model.

Updating the forward model is generally thought to be done by decreasing the prediction error over time (error-based learning) [24, 25]. Dominant theories in the field of motor control suggest that error correction is an optimal process [26, 27]. The amount of error correction depends on the reliability of the predicted sensory consequences and the reliability of the received sensory consequences [28]. However, in situations in which error feedback is unavailable or uninformative, the environment needs to be explored in order to find and exploit the most beneficial solution. This type of process is called reinforcement learning [29, 30]. It can also be beneficial in situations where the average end-point error is zero, but the movement is still updated in order to become more efficient [31]. Reinforcement learning, specifically exploration, usually leads to enhanced variability in the explored dimension of the behavior. Furthermore, research suggest that the nervous system could actively regulate this process when needed [32]. For this reason, we hereafter use exploration strategies to refer to the active process of learning through exploration. Some studies have found that reinforcement learning could also accelerate learning in error-based learning tasks [32]. More specifically, a benefit of task-specific baseline variability in adaptation during spatial perturbation paradigms has been found. This means that even if sensory feedback is available to the participant, more variability in the task-specific dimension could facilitate learning. It is thought that the brain has a memory of previously encountered errors [33]. Similarly, it has been proposed that predicted (future) errors might activate learned weights for motor primitives (neural mechanism that coordinate a set of movements rather than independent movements), which could elevate learning [34]. In line with this idea, reinforcement learning could lead to more experience with different movement errors and facilitate correcting future errors [35]. However, the idea that increased variability leads to better adaptation has been questioned by other studies claiming that variability originates from the different types of noise coming from motor control processes, which can have either positive, neutral or negative effects on adaptation [13].

Thus, the role of variability on adaptation is still under debate. The main question remains, can variability caused by exploration strategies directly benefit the updating of the forward model of a motor command, or are the two merely a result of error correction strategies? So far, the main focus of these studies has been the role of spatial and force-field perturbations. However, as large delays are inherent to the sensorimotor system and to contemporary electronic devices, temporal perturbation tasks have given great insights into adaptation in this domain [36–39]. It therefore is valuable to examine if task-specific variability in temporal features of movement can predict adaptation to these delays. Temporal perturbations (delays)

require humans to identify a new temporal relationship between the motor command and the sensory consequences. The effect of delays on the spatial prediction error is increased with higher movement speeds (i.e. moving faster increases the gap between hand and delayed feedback), while lower speeds have less effect on the size of the prediction error. Variability in movement speed can therefore help to explore this temporal relationship, and benefit the examination of newly introduced delays. If exploration itself provides a benefit in adaptation, we do not expect the movement speed to be systematically related to previously encountered errors. However, a recent study by [40] found that error correction strategies could explain differences of the learning rate in a balancing task. If error correction rather than exploration is the main benefit for adaptation, higher rates of error correction are to be expected throughout the baseline for people that adapt more. The aim of the study was to examine if higher baseline movement speed variability can predict adaptation in a temporal timing task. As we found a positive relationship, we followed up this analysis by examining if this relationship was the result of exploratory behavior, or if higher variability might be the result of increased error correction.

## Materials and methods

### Participants

20 students of the university of Barcelona (20.3 ±SD 2.3 years; 15 female) participated in the experiment. All participants gave written consent. In order to be eligible for inclusion, participants' vision and hearing needed to be normal or corrected to normal, and they needed to be free from movement restrictions or problems. The study was part of a program that has been approved by The University of Barcelona Bioethics Committee (CBUB) (Insitutional review board IRB 00003099) and was conducted in accordance to the Declaration of Helsinki.

### Apparatus

The setup of the experiment is shown in Fig 1A. Participants were seated in front of a graphic tablet (Calcomp Drawing Tablet III 24240), which recorded the movements of a hand-held stylus that the participants used in the experiments. A half-silvered upward facing mirror was located above the graphic tablet. Above the mirror, a projector displayed an image of the task onto a horizontal back-projection screen, at a frame rate of 72 Hz and a resolution of 800 by 600 pixels. The participant could see the reflection of this image in the mirror. The hand and the hand-held stylus were blocked from view by the mirror. A Macintosh Pro 2.6-GHz Quad-Core computer recorded the position of the stylus at 125 Hz and controlled the projected image. We calibrated the setup by aligning the position of the stylus with the location of five projected dots. This calibration allowed us to accurately display a cursor that represented the location of the stylus above the graphic tablet. The projected background was black (depicted as gray in Fig 1A). The feedback had a systematic delay of approximately 40 ms with relation to the hand movement. This value was determined before the start of the study with an interception task (similar to the one described below), in which we compared the stylus position and the target location on trials in which participants indicated they had hit the target. The system delay will have been corrected for in all of the future display and analysis of the data.

### Task

Fig 1 shows the setup and the procedure for the experiment. The goal of the task was to intercept a target (white dot) on the screen with the cursor (red dot), that followed the movement of the hand. In order to start a trial, the participant had to move the cursor to the starting

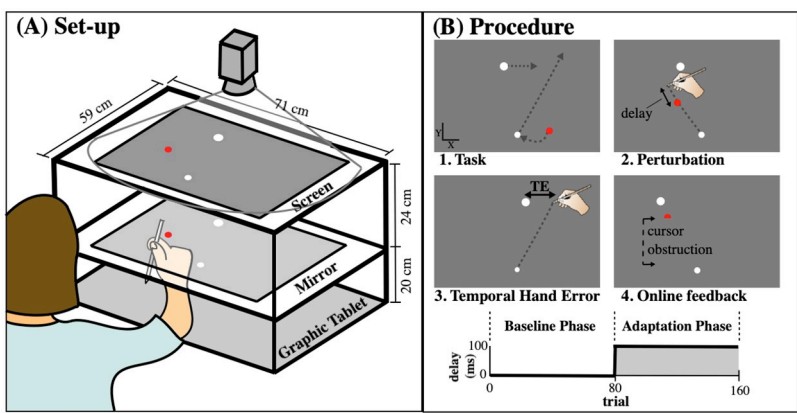

**Fig 1. Set-up and procedure.** A. Set-up of the task: The task image was projected onto a screen that was visible for the subject through a mirror. Recorded movements with a stylus were made on a graphic tablet. The hand and hand-held stylus were blocked from view by the mirror. B. Task procedure: (1) A white dot at the bottom of the screen (diameter = 6 mm) indicated the start location. A red cursor represented the movements made by the stylus (diameter = 6 mm). The trial initiated when the participant moved the cursor to the start location. The goal of the task was to intercept a ball (diameter = 10 mm) moving from left to right or vice versa with a straight ballistic movement. (2) During the adaptation phase there was a temporal perturbation (delay) of 100 ms between the hand and cursor. (3) The Temporal Hand Error (TE) was defined as the temporal lead or lag of the stylus (hand) crossing the target's trajectory line in relation to the target. A lead was denoted by a positive TE, and a lag with a negative TE. (4) In the area within the dashed lines, vision of the cursor was obstructed from view.

position (Fig 1B-1). A high-pitched sound marked the start of the trial. Randomly across trials, the target would move from left to right or vice versa at one of 3 different speeds (17.5, 22.5 or 27.5 cm/s). The three different speeds were used to prevent participants from learning were to intercept at what time, without taking the movement of the target into account. The target path was +20 cm in the y-direction from the starting position. Participants were instructed to intercept the target with a smooth straight movement, and to end their movement well beyond the target's trajectory line. There were no points rewarded, only when the center of the cursor hit the target, an acoustic signal would sound.

After the explanation of the task and a familiarization phase of 10 trials, the experiment started. The experiment consisted of a baseline phase followed by an adaptation phase. During the baseline phase of 80 trials, no additional delays were added to the system. During the adaptation phase, the cursor movement was delayed with an additional 100 ms compared to the hand movement (Fig 1B-2). As participants could not see their own hand, they were unaware of these additional delays. This was verified by verbal confirmation at the end of the experiment.

To discourage online corrections we blocked the cursor's path from view between 0.5 and 16 cm (see Fig 1B-4) along the y-axis from the starting location. During the baseline this led to an average of 126 ms (95% bootstrapped CI [124-128 ms]) visibility of the cursor before it would reach the trajectory line. As a result, participants could see the cursor cross the target's trajectory line, but online corrections were minimized.

## Analysis of responses

The data was prepared for analysis with the aid of the R-program [41]. In order to account for any deficient recordings and missing data, the raw position data for the hand and cursor in the x and y direction were interpolated by steps of 0.008 s and filtered with a bidirectional Butterworth filter (cut-off = 6Hz, sampling rate = 125 Hz). The velocity and acceleration of the hand were computed by calculating the first and second order derivatives of the filtered position

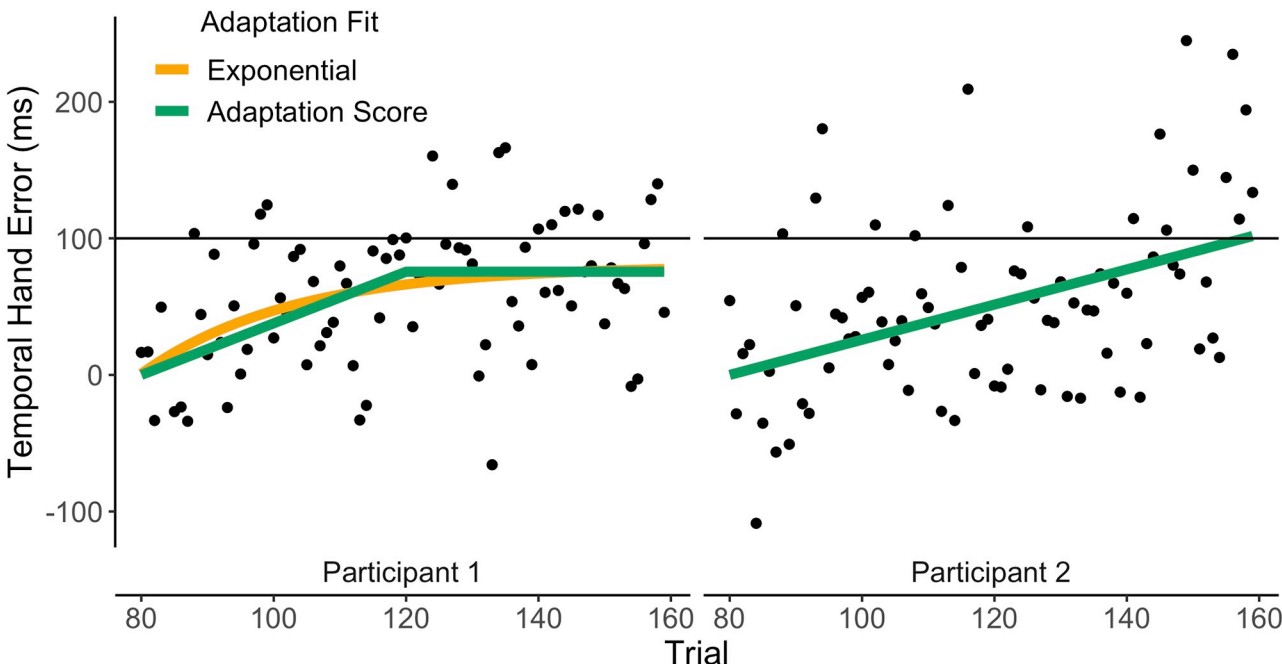

**Fig 2. Adaptation score examples.** The dots represent the moving averages (window = 4 trials) of two example participants. In order to calculate the Adaptation Score, the slope (a) and asymptote (b) were normalized across subjects and then summed for each participant. For P1 we were able to fit an exponential function, while for P2 the adaptation an exponential could not be fit.

data with respect to time. The derivatives were calculated for each time point by dividing the change in position by the change in time over the past two time points (average $\Delta t = 0.016s$).

Of the 20 participants, one was excluded from the study due to recording issues with the graphic tablet. For the remaining 19 participants (3040 trials), we inspected the movement speed during the movement before the target's trajectory line was reached. As we had instructed participants to move through the target's trajectory line, we discarded trials in which subjects moved through it with very low speeds ($v < 20$ cm/s). Two other participants were removed from the study because the systematically failed to comply with these instructions of the task (in >25% of their trials). For the 17 included participants we removed the trials in which this was the case (2.1% of the remaining trials discarded).

We calculated the Temporal Hand Error (TE) on each trial. TE was defined as the temporal difference between the hand and the target reaching the same point on the target's trajectory line (Fig 1B-3). The TE was baseline-corrected by subtracting the average TE of the last 30 baseline trials for each participant in order to make sure any biases within subjects were removed. Positive values denoted the hand leading the target and negative values denoted the hand lagging behind the target. In order to hit the target when a temporal perturbation was applied, subjects would have to start leading the target with their hand. The TE was used to calculate the Adaptation Score for each participant. The Adaptation Score was calculated by fitting a two-state line through the TE data points of the adaptation phase (see Fig 2), with intercept 0. The slope (a) and asymptote (b) that resulted in a minimum residuals were each first normalized across subjects and then summed. The Adaptation Score gave us the ability to reward both speed and final level of adaptation. High values of the score meant that participants were relatively fast and had more complete levels of adaptation. Low values meant that participants adapted slowly and incomplete. Although we had the possibility to analyze slope and asymptote separately (to see if speed or final level of adaptation was affected differently)

we only had a few participants with higher slopes, which made this analysis more difficult (see Table 1). We therefore decided only to focus on the Adaptation Score.

Adaptation is often quantified by fitting the exponential function ($exp(-\tau \cdot trial)$) and calculating the time constant $\tau$ of this function. However, our pilot data showed we could not fit this function for each participant, as some participants adapted in a non-exponential way. There were 11 participants for which we were able to calculate the exponential function. In order to verify our method of quantifying adaptation, we compared $\tau$ of this exponential to our Adaptation Score and confirmed our main finding (see below) using $\tau$ instead of the Adaptation Score.

Movement onset was calculated according to algorithm A as proposed by [42] on the tangential velocity of the hand (tolerance range = 10%). The average movement speed (Mv) between the time of movement onset and the crossing of the target's trajectory line was calculated. We also calculated the absolute movement angle (Ma) from the y-axis, calculated between movement onset and movement onset + 200 ms.

In order to identify the relationship between different kinematic variables and Temporal Hand Error, we analyzed the relationship between Mv, Ma, movement time (T), absolute target velocity (tv), $(\sigma^{Mv})^2$- group (LOW = 0; HIGH = 1) and their interactions on TE. We did this with a LASSO Regression for both the baseline and adaptation phase (*lambda* = 0.6). LASSO Regression can be used with multicollinear variables. We calculated bootstrapped confidence intervals (95%). LASSO regression includes a penalization. Confidence intervals can therefore only provide information about the variability regarding $\beta$ for each predictor, but they should be interpreted with caution when it comes to the significance of the predictors.

Additionally, we performed another analysis with this technique in order to verify the respective contributions of movement speed variability ($(\sigma^{Mv})^2$) and movement angle ($(\sigma^{Ma})^2$) and their interaction on the Adaptation Score. As expected with regards to our hypothesis, we found the only included predictor of the Adaptation Score to be the $(\sigma^{Mv})^2$ (see Results). The remainder of our analysis was therefore focused on the role of Mv on Adaptation.

**Table 1. Pre-normalized slopes (a) and Asympotes (b) and adaptation scores for all the subjects.**

| a (unnormalized) | b (unnormalized) | Adaptation Score |
|---|---|---|
| 7.1 | 86 | -0.298 |
| 3.3 | 70 | -1.49 |
| 8.8 | 70 | -1.03 |
| 1.1 | 110 | 0.503 |
| 7.9 | 64 | -1.42 |
| 34.3 | 76 | 1.33 |
| 1.2 | 100 | 0.266 |
| 1.3 | 112 | 0.684 |
| 3.7 | 75 | 1.49 |
| 7.5 | 82 | -0.465 |
| 7.3 | 73 | -0.991 |
| 3.4 | 72 | 1.08 |
| 1.3 | 113 | 1.67 |
| 1.2 | 110 | 0.611 |
| 1.9 | 76 | -1.28 |
| 2.7 | 77 | -1.15 |
| 0.90 | 109 | 0.492 |

The correlation between Mv variability across baseline trials $((\sigma^{Mv})^2)$ and the Adaptation Score was calculated using the rcorr- function from the Hmisc-package [43]. We used Spearman, in addition to Pearson, because Spearman (based on ranks) captures monotonic relation better than Pearson (which requires linear relations). We also calculated this relationship for $\tau$ calculated for the 11 participants that showed exponential adaptation curves.

We examined if the predictive capacity of $(\sigma^{Mv})^2$ was due to the sequential effects of the TE on Mv by dividing the participants in a LOW (n = 9) and HIGH (n = 8) $(\sigma^{Mv})^2$ group based on the size of their movement speed variability. In order to verify if there were group differences in variances for Mv and TE, we did F-tests on the results from the baseline phase (var.test-function for R). As we suspected that the benefit of $(\sigma^{Mv})^2$ came from error correction rather than exploration, we expected sequential effects of $TE_t$ on $Mv_{t+1}$. The sequential effects can show us if the error on one trial had an effect on the movement speed of consecutive trials. We verified this idea in two different ways: First, we fit a linear mixed model (lmer-function of the lme4-package [44] that examined the slopes of the HIGH and LOW $(\sigma^{Mv})^2$ group, with subject as a nested random effect. We found significant effect for the HIGH $(\sigma^{Mv})^2$ group, indicating that movement speed was used to account for previous encountered errors. In order to examine this effect in more detail, we calculated the cross correlation (ccf-function for R) between the TE and Mv for lag 0 through 3 for each subject individually to capture the corrective pattern. This way we could examine the effects of the movement speed on the error (lag 0) and the effect of the error on the movement speed over the next 3 trials (lag 1-3). In order to decrease the effect of any auto-correlation or common trends within the time-series we prewhitened the TE and Mv for each participant with the prewhiten-function of the psd-package. Bootstrapped confidence intervals (95%) were calculated. The results were correlated with the individual Adaptation Score results, in order to examine if sequential effects of TE on Mv could predict differences in adaptation.

## Results

We first analyzed the behavior or participants within a trial for both phase 1 and 2 (see Table 2). A negative TE denoted being late, and a positive TE denoted being early. Both phases

**Table 2. Predictors for temporal hand error within a trial for Phase 1 and 2.** λ = 0.6 for both models. As LASSO regression includes a penalization, caution is required when interpreting Confidence Intervals.

| Predictor | Phase 1 $\beta$ [95%CI] | Phase 2 $\beta$ [95%CI] |
|---|---|---|
| Ma | . | . |
| Mv | . | . |
| tv | . | . |
| HIGH $(\sigma^{Mv})^2$ | 4.74 [0.24–9.48] | 13.44 [10.55–26.88] |
| T | -18.36[-36.72–(-5.52)] | -1.87[-11.68–7.43] |
| Ma · Mv | -0.0016 [-0.0060–(-0.00073)] | -0.011 [-0.021–(-0.0072)] |
| Ma · tv | -0.0026 [-0.0087–0.0049] | . |
| Ma · HIGH $(\sigma^{Mv})^2$ | . | . |
| Ma · T | 0.40 [0.090–1.04] | 2.15 [1.38–4.30] |
| Mv · tv | . | . |
| Mv · HIGH $(\sigma^{Mv})^2$ | 0.035[-0.156–0.070] | . |
| Mv · T | -1.67 [-3.09–(-0.69)] | -1.28 [-2.57–(-0.72)] |
| tv · HIGH $(\sigma^{Mv})^2$ | 0.15 [0.02–0.35] | 0.35[0.20–0.70] |
| tv · T | -0.47 [-0.93–(-0.34)] | -0.70[-1.25–(-0.58)] |
| HIGH $(\sigma^{Mv})^2$ · T | . | . |

included $(\sigma^{Mv})^2$-group and movement time (T) as predictors of TE. Larger movement times led to a more negative TE. The HIGH $(\sigma^{Mv})^2$ group had on average more positive TE. None of the other main effects were included as predictors, although there seemed to be some small interaction effects. As movement time was one of the main predictors of error, it is likely that participants use Mv to account for errors. However, another way to account for errors is by varying the movement angle (Ma). We found a significant correlation between Mv and Ma (R = 0.62, p < 0.001), and $(\sigma^{Mv})^2$ and $(\sigma^{Ma})^2$ (R = 0.67, p < 0.005). The results of the LASSO Regression ($\lambda$ = 0.37) showed us that only $(\sigma^{Mv})^2$ was a significant predictor of the Adaptation Score ($\beta$ = 0.01; 95% CI[0.008–0.020]), while $(\sigma^{Ma})^2$ and $(\sigma^{Ma})^2 \cdot (\sigma^{Mv})^2$ were dropped. The remainder of the analysis will therefore only focus on Mv.

The experiment was designed to study the relationship between $(\sigma^{Mv})^2$ and the Adaptation score. Fig 2 shows two example subjects and their responses during the experiment. Participants showed various degrees of adaptation. Adaptation scores varied between -1.49 and +1.67 (Table 1).

Fig 3A displays the moving average (window = 4 trials) of TE throughout the task. The blue line shows the mean of the nine participants with below-average $(\sigma^{Mv})^2$, and the orange line depicts the mean of the eight participants with above-average $(\sigma^{Mv})^2$. The spread around these lines represent the standard deviation of the data across subjects. When the temporal perturbation of 100 ms is introduced, subjects account for this delay by reaching the target's trajectory earlier with their hand in relation to the target. This is depicted by a more positive TE in Fig 3. Upon visual inspection there seemed to be a slightly faster adaptation for the HIGH group than for the LOW group (see Fig 3B). In order to verify this notion, we calculated an adaptation score for each participant. We found a significant correlation between $(\sigma^{Mv})^2$ and the adaptation score (Pearson r = 0.51, p = 0.037). We found an even higher Spearman correlation (r = 0.61, p = 0.009), indicating that the relationship might be better described by a logarithmic

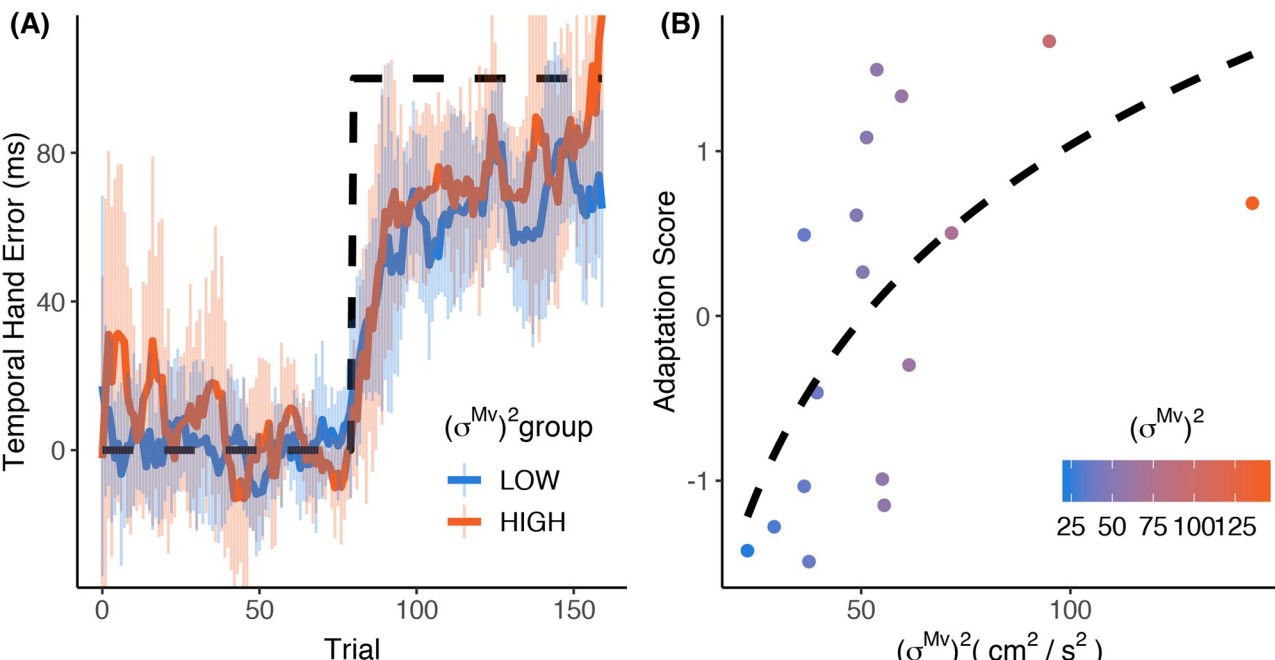

**Fig 3. The relationship between $(\sigma^{Mv})^2$ and adaptation score.** (A) The moving average data (+- SD) of the LOW (blue) and HIGH (orange) variability group. (B) The relationship between $(\sigma^{Mv})^2$ and the Adaptation Score.

curve (dashed line in Fig 3B). These findings imply that there is a strong positive relationship between $(\sigma^{Mv})^2$ and the adaptation. We confirmed this result by finding a negative correlation between $(\sigma^{Mv})^2$ and $\tau$ (r = -0.63, p = 0.038) (Fig 4A). This was as expected, as the Adaptation Score had a negative (albeit not significant) correlation with $\tau$ as calculated by the exponential (r = -0.54, p = 0.088) (Fig 4B).

As expected, the F-test revealed a significant difference between the HIGH and LOW variability groups in baseline Mv (F(1, 1315) = 2.5, $p < 0.001$). However, no differences were found for TE (F(1,1315) = 1.1, p = 0.35). This is important because it ensures that differences in sequential effects that are found between groups are not the result of differences in errors that were made.

We examined the relationship between the TE on a certain trial, and the change in Mv on the next trial for participants in the HIGH and in the LOW $(\sigma^{Mv})^2$ group (see Fig 5). The mixed linear model revealed that, when the error was zero, neither of the groups significantly changed their Mv (p = 0.69 (LOW); p = 0.59 (HIGH)). The effect of TE on the next trial's Mv was significant for the HIGH $(\sigma^{Mv})^2$ group (slope = -15.9 $cm/s^2$; p = 0.02). The LOW $(\sigma^{Mv})^2$ group did not show a significant effect (slope = 6.3 $cm/s^2$; p = 0.68). The ANOVA revealed a significant difference between groups (F(1, 1263) = 5.2, p = 0.02). This suggests that participants with a higher $(\sigma^{Mv})^2$ increased their speed when encountering a negative error and vice versa, while the participants with lower $(\sigma^{Mv})^2$ did not.

In order to examine if this difference can predict adaptation on an individual level, we calculated the cross correlation (ccf) between Mv at different t+lag and TE (Fig 6). A negative ccf indicates that a negative error (arriving too late) leads to increased velocities, and a more positive error (earlier) leads to decreased velocities. Consequently, a positive ccf indicates the opposite (negative error → decreased velocities; positive error → increased velocities). In congruence with Fig 6, participants with higher $(\sigma^{Mv})^2$ scores seemed to have a trend towards

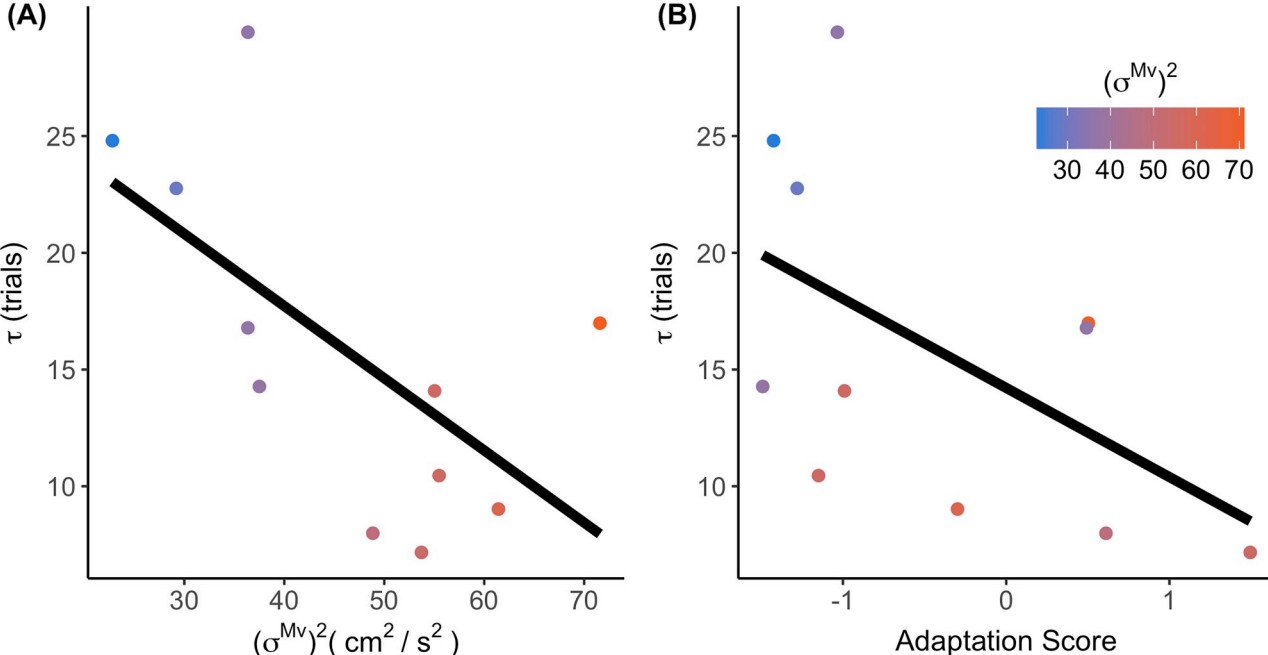

**Fig 4. Verification of the adaptation score.** A. The relation between movement speed variability and $\tau$ of the exponential function. B. The relationship between the Adaptation Score and $\tau$ of the exponential function.

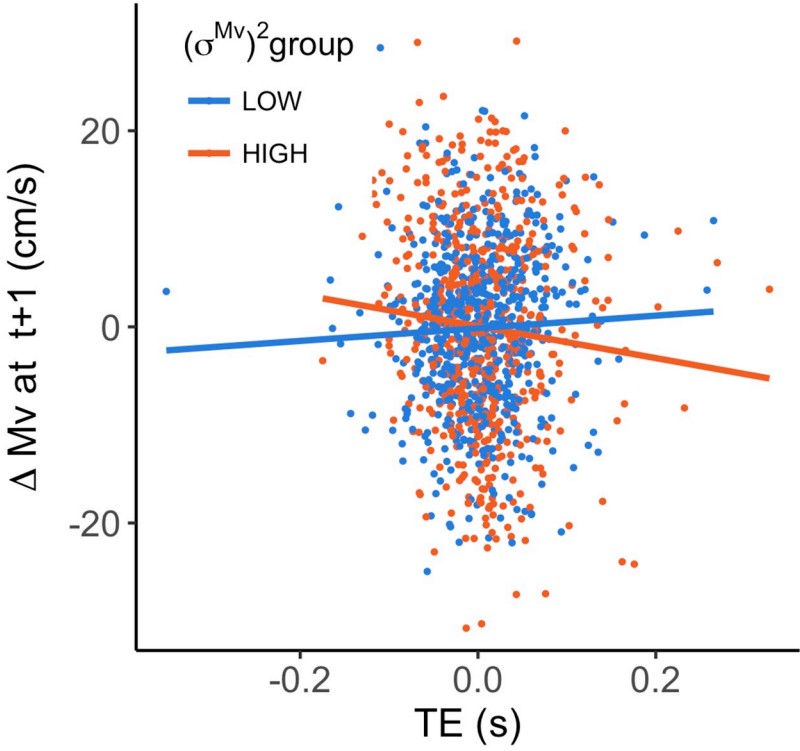

**Fig 5. The relationship between the baseline TE on trial t-1 and the subsequent change in Mvel.** Dots represent individual trials for subjects within the HIGH (orange) and LOW (blue) $(\sigma^{Mv})^2$ group. The lines indicate the predicted slope of the model.

more negative ccfs, while the participants with low $(\sigma^{Mv})^2$ scores had ccfs near zero. The high $(\sigma^{Mv})^2$ group showed a positive ccf(0) (i.e. the cross correlation between Mv and TE within a trial). Slower hand movements led to more negative TE and fast hand movement to positive TE. It shows that for this group there is some dependency of the TE on the Mv within a trial.

To see if this difference could explain the differences in adaptation, we calculated the correlation between the different lags and the Adaptation Score (Fig 7). The correlation between Mv and TE within a trial (lag0), i.e. how the Mv affected the TE, seemed to not be able to explain differences in adaptation. Although there is also no correlation for ccf(1), the lag2 and lag3 ccf seem to predict the Adaptation Score in this experiment. This suggests that errors might not be corrected right away, or only partially from trial to trial.

## Discussion

This study found a positive relationship between variability of movement speed during the baseline phase, and the adaptation to a temporal perturbation. Our results indicate that this variability might be a result of error correction strategies, rather than exploration. Participants that corrected their movement velocity after encountering an error on average adapted more than participants who did not. To our knowledge, we are the first ones to reveal such a relationship in temporal adaptation.

The role of noise in adaptation has recently received a lot of attention. A study by Wu et al. [32] suggested that there might be a positive influence of variability on adaptation through reinforcement learning, revealed by an increased baseline variability for fast adapters. In line with these results, we found a benefit of increased baseline variability on the adaptation for

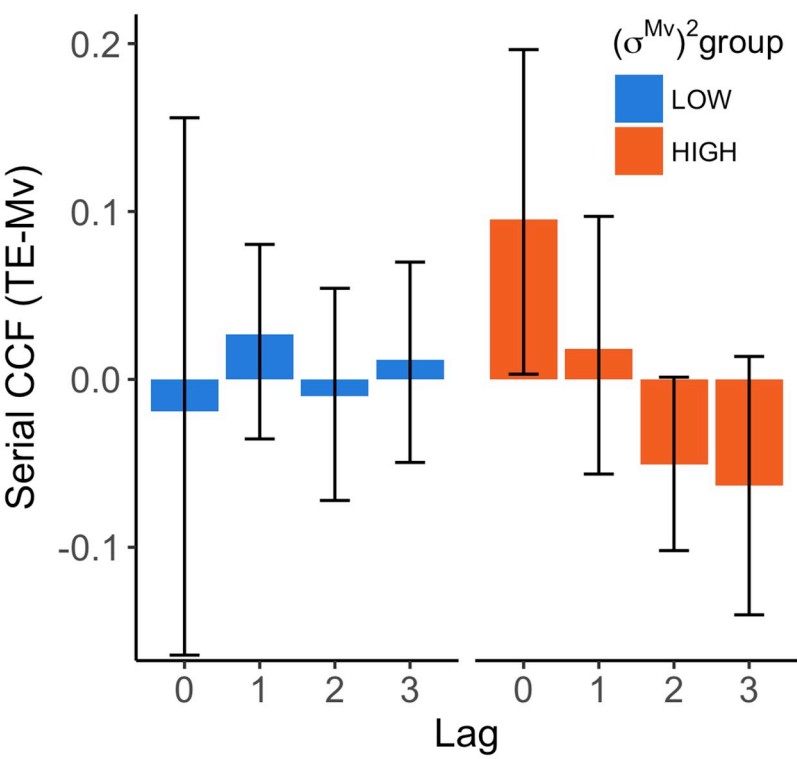

**Fig 6. Serial ccf for each $(\sigma^{Mv})^2$ group.** The serial ccf for lag 0-3 for the HIGH (orange) and LOW (blue) $(\sigma^{Mv})^2$ groups. The error bars represent the 95% bootstrapped confidence intervals.

temporal perturbations. However, our results suggest that participants with higher variability use their velocity more as a way to correct for errors. Our results are therefore more in line with the idea that motor noise from different types of processes can have either a positive, negative or neutral effect on adaptation [13]. Two types of noise that are relevant for adaptation are sensory noise and motor noise. Sensory noise can influence adaptation by influencing the uncertainty of the error [22, 45]. More sensory noise can lead to higher variability. However, error correction actually decreases with more uncertain sensory information. It is therefore unlikely that sensory noise was the source of the increased adaptation. Motor noise can roughly be divided into planning noise and execution noise [16]. A recent study by Vliet et al. [46] found that execution noise correlated negatively with adaptation rate, while planning noise revealed the opposite trend. Execution noise increases the uncertainty of the feedback that is received. Contrarily, more noise in the planning process can lead to more uncertain forward predictions, with a higher rate of error correction as a result. In the current study, we did not measure noise in the system directly. We therefore cannot determine the direct effect of planning noise and/or execution noise on the adaptation. However, it has been suggested that planning noise, in the form of stochastic resonance, could have a positive effect on signal detection by enhancing sub-threshold signals to supra-threshold [47–49]. Enhanced error-detection could lead to more error correction. Our results are in line with the idea that higher planning noise could increase the reliance on error feedback and lead to larger trial-to-trial corrections [46].

The question remains: why did we find a positive relationship between baseline variability and adaptation, while many other studies did not [13, 50–53]? It is likely that differences in design, such as variability measure or feedback provided, trigger variations in the

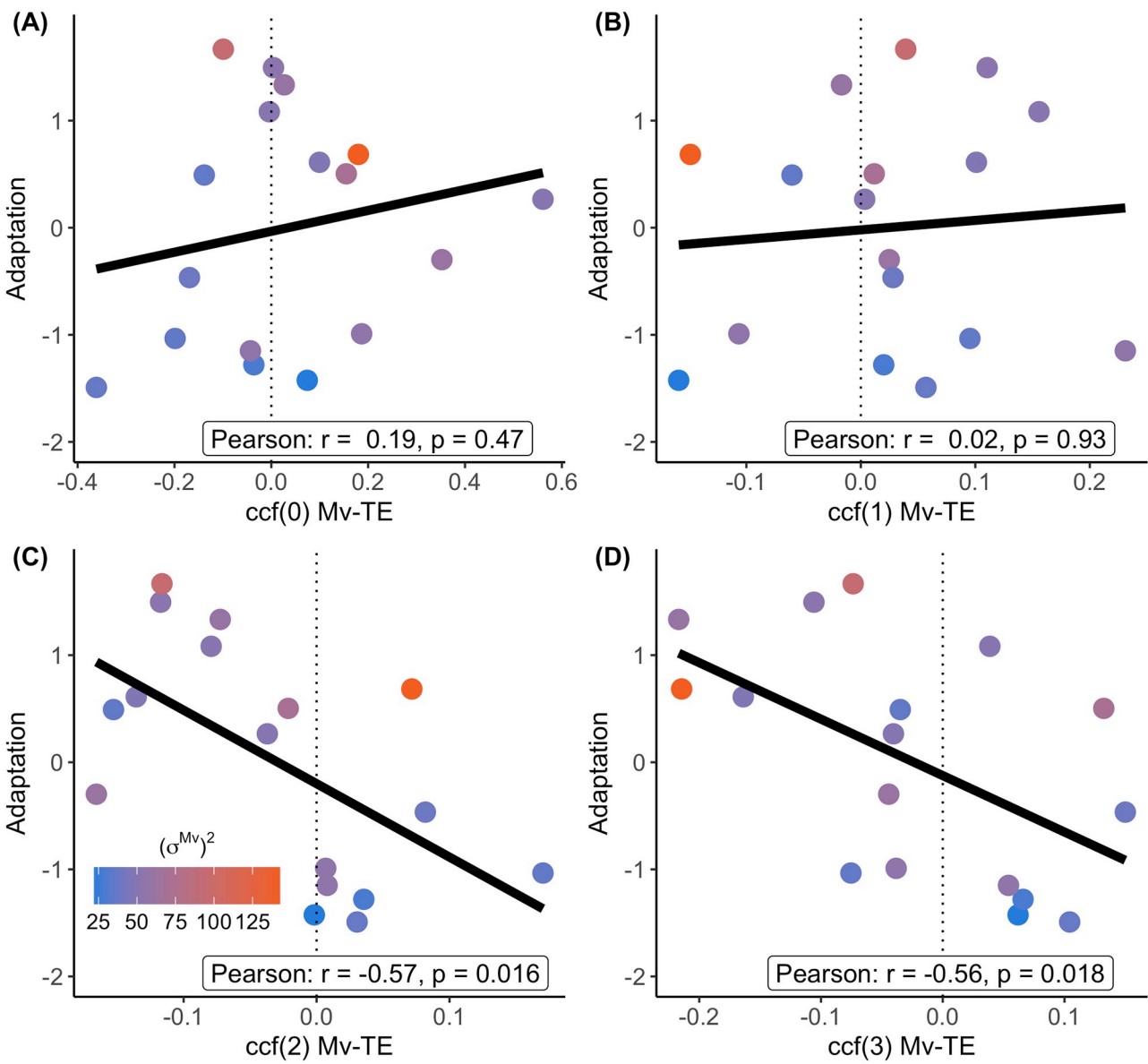

**Fig 7. The relationship between the ccf score and the adaptation score.** The ccf is shown for lags 0-3 (A-D). The color of the data-points (blue-orange) indicates the size of the $(\sigma^{Mv})^2$. The dotted line indicates a ccf of zero.

accumulation of the different types of noise. In this study, we defined the average movement speed as an indicator of task-relevant variability, instead of the movement speed at the moment of interception, or the error at interception. We theorized that movement velocity would provide the richest information about the temporal contingencies of the sensorimotor system, and therefore could be more beneficial for exploration. By averaging the movement speed on each trial we believe to have filtered out a large part of the execution noise. This could have highlighted the effect of planning noise on adaptation. Similarly, Wu et al. [32] calculated the variability of deflection of a rewarded hand path throughout the movement, while He et al. [13] looked at the endpoint error. It is likely that endpoint error is more susceptible to execution noise. On the other hand, Dhawale et al. [54] suggested in a recent review that

differences in results between studies might be inherent to differences in feedback. In Wu et al. [32], feedback was absent during the baseline, which may have led to an accumulation of planning noise (drift) that could not be corrected trial-to-trial [55]. In He et al. [13], feedback was provided throughout the movement, which means the variability measure might be dominated by execution noise. In the current study, we did provide error feedback in the baseline, which might be taken as evidence against the latter idea. However, these ideas together support the hypothesis that experimental designs that promote planning noise over execution noise are more likely to reveal a positive relationship between variability and adaptation.

Next, we need to consider the effect of generalization on the results. The term generalization refers to how the mapping between action and consequence generalizes to other tasks and target speeds. Research has shown that generalization of temporal perturbations is limited to tasks that are very similar to the learned task [39]. We used three different target speeds in two directions in this study. Some participants might had a better generalization from one speed to the other than others. It is possible that at the moment the perturbation was applied, some participants were able to generalize the changing consequences better for all target speeds and, as a result, adapt faster. For generalization to have an effect on our results there would need to be a relationship between movement speed variability and generalization. More variable people might develop broader tuning curves, and as a result adapt faster. We think this idea is captured well within the notion of exploration that is part of our original hypothesis: higher variability due to exploration might broaden the brain's knowledge on the temporal contingencies over a broader range of target speeds and situations. Unfortunately our current dataset cannot give a definitive answer to the question of generalization.

Throughout the study, we mainly focused on how participants change their movement speed from trial-to-trial. However, in this task subjects were free to change their interception location or movement angle and onset time in order to account for prediction errors. This leads to a variety of strategies that participants can use to perform the task and correct for errors. These strategies are not independent of the target speed, nor independent of each other. For example, faster targets can lead to faster movements, earlier onset times and/or to broader hitting positions [56–58]. Furthermore, we found a correlation between the movement angle and the movement speed. These interconnections between movement features make it challenging to examine the participants' strategies. However, it has been proposed that, in order to successfully hit a target, people benefit more from estimating the time it takes to reach the target's trajectory, followed by fine-tuning the point of interception throughout the movement (online correction), as opposed to estimating when the target will reach the interception location [2]. As a result, errors are likely attributed to incorrect estimations of the time it takes to reach the target's trajectory, rather than an incorrect estimations of interception location. As online correction was discouraged in our experiment, the reliance on error correction through changing the predicted temporal features of the movement was enhanced even further. During this experiment, successful participants used their movement speed as a control agent for trial-to-trial error correction in interception, and to a lesser extend their interception location or movement angle. This does not mean that low adapters do not have useful strategies of error correction. However, if the source of an error is temporal, using movement speed as a corrective strategy can be beneficial. Vice versa, if an error is spatial, corrections in movement angle would be more favorable. It is therefore possible that the difference in adaptation is rooted in the interpretation of the error.

Ccf(0) seemed to be slightly higher in the high variability group compared to the low variability group. This shows that participants with higher variability had a higher cross-correlation between the speed of movement and the temporal hand error in the same trial. This suggests that the temporal hand errors that participants with higher variability make are more

dependent on the movement speed. This is not a surprising feature, given that more variability can more easily reveal a dependency of the temporal hand error on the movement speed. Another reason why ccf(0) is higher in the HIGH variability group is that participants with more dependency of temporal hand error on movement speed use their movement speed more as a way to correct for error and as a result, adapt more. We need to consider the possibility that this positive ccf(0) could have had an effect on the negative ccfs found in later lags. When a slow hand movement is made and a negative error occurs, it is more likely that the next movement will be faster. This therefore can lead to negative ccfs in consecutive lags. However, these differences in ccf(0) were not able to explain differences in adaptation. Therefore it seems unlikely that dependencies between Mv and TE within a trial alone could have explained more negative ccfs for consecutive lags.

To conclude, we found a correlation between movement speed variability and adaptation to a temporal perturbation. Further analysis of this relationship indicated that this is likely the result of error correction strategies that also benefited the adaptation, although the results are not conclusive. More research is needed to examine the contributions of movement speed, interception location and movement angle and the differences between temporal and spatial perturbations. Furthermore, it would be interesting to know how these error correction strategies are related to different types of noise and how the how the brain regulates these processes.

## Acknowledgments

We would like to thank Cristina de la Malla and Björn Jörges for their stimulating discussion and help evaluating the manuscript.

## Author Contributions

**Conceptualization:** Elisabeth B. Knelange, Joan López-Moliner.

**Formal analysis:** Elisabeth B. Knelange.

**Funding acquisition:** Joan López-Moliner.

**Investigation:** Elisabeth B. Knelange.

**Methodology:** Elisabeth B. Knelange, Joan López-Moliner.

**Project administration:** Joan López-Moliner.

**Resources:** Elisabeth B. Knelange, Joan López-Moliner.

**Software:** Elisabeth B. Knelange, Joan López-Moliner.

**Supervision:** Joan López-Moliner.

**Visualization:** Elisabeth B. Knelange.

**Writing – original draft:** Elisabeth B. Knelange.

**Writing – review & editing:** Elisabeth B. Knelange, Joan López-Moliner.

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
