## [Decision Letter · Decision Letter 0]

7 Aug 2019

PONE-D-19-18039

Increased error-correction leads to both higher levels of variability and adaptation

PLOS ONE

Dear Dr. López-Moliner,

Thank you for submitting your manuscript to PLOS ONE. After careful consideration, we feel that it has merit but does not fully meet PLOS ONE’s publication criteria as it currently stands. Therefore, we invite you to submit a revised version of the manuscript that addresses the points raised during the review process.

The manuscript has been reviewed by two expert academic reviewers in your area of research. I share their assessment that this is an interesting topic of investigation looking at how individual differences in people's behaviour may predict later adaptation. Although I do invite a revision of the submitted work, the reviewers also highlighted significant issues with data analysis that required considerable work. I hope you may be able to address the reviewers concerns with data analysis convincingly. 

We would appreciate receiving your revised manuscript by Sep 21 2019 11:59PM. To enhance the reproducibility of your results, we recommend that if applicable you deposit your laboratory protocols in protocols.io, where a protocol can be assigned its own identifier (DOI) such that it can be cited independently in the future. For instructions see: http://journals.plos.org/plosone/s/submission-guidelines#loc-laboratory-protocols

We look forward to receiving your revised manuscript.

Kind regards,

Welber Marinovic

Academic Editor

PLOS ONE

Journal Requirements:

1. Thank you for including your ethics statement:  All participants gave written consent.  Their vision and hearing were normal or corrected to normal, and no movement  restrictions or problems were reported. The study was part of a program that has been  approved by the local ethics committee and was conducted in accordance to the  Declaration of Helsinki.

Please amend your current ethics statement to include the full name of the ethics committee/institutional review board(s) that approved your specific study

2. Please provide more information regarding the setting (e.g. locations, relevant dates, periods of recruitment, data collection) as well as the sources, methods and criteria of participants' selection.

Reviewers' comments:

Reviewer's Responses to Questions

**Comments to the Author**

1. Is the manuscript technically sound, and do the data support the conclusions?

Reviewer #1: No

Reviewer #2: Yes

2. Has the statistical analysis been performed appropriately and rigorously? 

Reviewer #1: No

Reviewer #2: No

3. Have the authors made all data underlying the findings in their manuscript fully available?

Reviewer #1: Yes

Reviewer #2: Yes

4. Is the manuscript presented in an intelligible fashion and written in standard English?

Reviewer #1: Yes

Reviewer #2: Yes

5. Review Comments to the Author

Reviewer #1: The authors investigated how movement variability is related to motor adaptation in the temporal domain. Although the motivation of the study and the target-intersection task sound interesting, I cannot agree with the validity of the analysis in the current paper. Please respond to the following questions carefully.

In sum, I was interested in the target-intersecting task, but I cannot be favorable for almost all the results.

Major

1) Even under assuming straight pen trajectories, this task is redundant. That is, subjects can choose the pair of initial movement angle and movement velocity to achieve the task. The authors should investigate how the pair is modified depending on perturbation and task set; however, they investigated only movement velocity. These missing analyses can hide significant findings inherent in this interesting experiment. Without the analysis of the movement angle, it is not worth to discuss their results.

2) In the adaptation, we can assume a simple state-space model, Mv(t+1) = lambda*Mv(t) + eta*(TE(t)+zeta(t)) + xi(t), where lambda is forgetting rate, eta is learning rate, xi(t) is motor noise, and zeta(t) is sensory noise to perceive TE(t). Under this equation, it is self-evident that the larger sigma_MV results in faster learning, because a high magnitude of xi(t)+eta*zeta(t) (i.e., sigma_MV) is nearly equivalent to high learning rate. Please mention clearly that the current results are not mere results of high sensory noise.

3) I wonder how generalization affected the results. Because the authors used three different speeds for target motions, they need to coednsider how the adaptation in one speed can be generalized to other speeds. Another possibility was the participants with small \\sigma_MV showed little generalization compared to those with large \\sigma_MV.

4) Please discuss slope and asymptote separately. Why did the authors sum those values to quantify Adaptation Score?

5) Despite the clarity of Fig. 3B (if the Adaptation Score were a valid value to quantify adaptation), Figs. 4-6 do not seem meaninful. The reason is the small correlation in Fig.3, cross-correlation that does not seem to be significantly different from 0 in Fig. 4, and low correlations in Fig.5. In Figs. 3 and 5, it seems impossible to predict one variable based on the other.

Reviewer #2: The authors examine the relationship between variability in hand movement speed during baseline and the adaptation to a temporal perturbation of sensory feedback (delay). They find that baseline variability is a good predictor of adaptation expressed as a summary score including the rate at asymptote level of adaptation. The authors argue that the increased baseline variability seen in ‘good adapters’ is not due to exploration, but rather to increased correction of errors in previous trials.

The source and relevance of motor variability, and the explanation of individual differences in motor learning, are critical issues in the field of motor control and learning. The authors provide important novel insights into the role of exploration and variability in motor learning. The experiment is well designed, and the paper is generally well written and concise.

My main comment is that the analyses of the relation between variability and adaptation should be more clearly explained and motivated. How/why were the particular analyses performed? Why was the adaptation score calculated this way? What is meant by sequential effects? How were the participants divided into groups? Why was the linear model fitted to the group data, and the cross-correlation performed per participant? Why are both a Pearson and a Spearman correlation reported; how exactly are variability and adaptation related? Why do some analyses test differences between groups, while other analyses focus on correlation of baseline temporal variability with other measures?

Another important point is that the term ‘exploration’ should be clearly defined. Is this an actively controlled (or deliberate) process, or an automatic, implicit process? The term ‘exploration strategies’ (used by the authors) makes me think that it is a deliberate process, but I am not sure this is what the authors mean.

Minor comments

Introduction

Lines 4-18. I understand what the authors are saying here but the reasoning is a little confusing. They first argue that we need sensory feedback, and then say that sensory feedback is too slow for corrections and we thus need an internal model. It might be helpful to distinguish between immediate corrections and adjustment to changes. In addition, we also need to predict the trajectory of the ball if we’re intercepting a moving object and rely on sensory feedback. Please clarify this paragraph.

L 22. “… neural variability”. If the authors refer to execution noise as well as planning noise, please remove ‘neural’.

L 30-31. “In situations in which error feedback is not clear…” I think ‘not available’ would be the correct term here. If not, please explain what is meant by ‘not clear’.

L 33-35. “It can also be beneficial …. in the explored dimension of the behaviour”. It’s not clear to me what this means, please expand.

L 36-38. Since the Wu et al paper is a main motivation for the current study, I think it would be helpful to expand the description of their findings a little bit.

L 42-45. It is not clear to me how exploration strategies could directly benefit the updating of the forward model, please clarify. And what is the potential role of variability from other sources than exploration?

Methods

Fig 1B. Display 4 is missing a title. It is unclear what the figure below the 4 displays belongs to.

L 92-93. What was the size of the target and the cursor?

L 109-110. 0.5 cm from where?

Fig 2 - legend. The word ‘summarized’ should be ‘summed’.

General. Were the results collapsed for the different target speeds?

Results

L 177. “There seems to be a slightly faster adaptation for the HIGH group than for the LOW group.” This is not clear to me from figure 3A. Was the slope or asymptote level of adaptation different between groups?

Fig 4. I would just like to note that I don’t find this figure particularly convincing. From the scatter, it looks like there is no relationship in either of the groups.

L 206-207. Shouldn’t the lag zero correlation be positive for both groups?

Fig 5. What do the error bars represent?

L 213-214. A word is missing in this sentence.

Discussion

Could the authors speculate about the origin of differences in error correction? For example, do people have differences in error sensitivity? Is the mechanism similar to differences in adaptation rate resulting from differences in error size or different levels of uncertainty of sensory feedback?

6. PLOS authors have the option to publish the peer review history of their article (what does this mean?). If published, this will include your full peer review and any attached files.

Reviewer #1: No

Reviewer #2: No

---

## [Author Response · Author response to Decision Letter 0]

23 Oct 2019

Dear editor and reviewers,

Thank you very much for spending time and effort improving this manuscript. It has been a very useful process. The response to the Reviewers PDF file contains all the answers and also some figures that cannot be included here. I refer you to this file for these figures.

Response to the reviewers:

The authors would like to thank the reviewers for their insightful comments on our manuscript. We have carefully evaluated each of their responses and concerns. We also thank the reviewers and editor for their patience and extending the deadline for our re-submission. Each of the comments will be evaluated separately below.

Reviewer 1

The authors investigated how movement variability is related to motor adaptation in the temporal domain. Although the motivation of the study and the target-intersection task sound interesting, I cannot agree with the validity of the analysis in the current paper. Please respond to the following questions carefully. In sum, I was interested in the target-intersecting task, but I cannot be favorable for almost all the results.

ANSWER: Dear reviewer 1, Thank you very much for your evaluation. The points that were brought up have proven very useful to us when improving our manuscript. We hope that we have clearly understood your concerns and that you are satisﬁed with our answers and corrections.

1. Even under assuming straight pen trajectories, this task is redundant. That is, subjects can choose the pair of initial movement angle and movement velocity to achieve the task. The authors should investigate how the pair is modiﬁed depending on perturbation and task set; however, they investigated only movement velocity. These missing analyses can hide signiﬁcant ﬁndings inherent in this interesting experiment. Without the analysis of the movement angle, it is not worth to discuss their results. 

ANSWER: We acknowledge that interception location plays a role when evaluating the movements of our participants. Variability in movement speed and aiming angle are correlated in our data (see ﬁgure 1 PDF). The reason why only movement speed was analyzed was because of the hypothesis that was the base of the research. As previous research had claimed exploration (and as a result variability) could beneﬁt error-based learning, our hypothesis followed this same rhetoric. We assumed that theoretically it would make more sense that exploration of movement speed would beneﬁt temporal adaptation over exploration in the movement angle, (and movement angle might beneﬁt spatial adaptation more). However, we realize that this assumption might blind us from other behavior that could have beneﬁted adaptation. In order to get more information on this idea we have modeled Partial Least Squares Regression (PLS) on the adaptation scores with absolute movement angle variability σ-Ma and movement speed variability σ-Mv as predictors. This PLS can give us information on the contribution of diﬀerent correlated predictors on a dependent variable.

AdaptationScore = 1.27 · σMv + 0.56 · σMa − 1.33 · (σMv · σMa )

This relationship in itself is very interesting, because together a larger part of the Adaptation Score can be explained than by just the σ-Mv. It indicates that both σ-Mv and σ-Ma have a positive main eﬀect on the Adaptation Score, but that either high or low values in both negatively aﬀects this relationship. However, of these predictors, only for σ-Mv the estimated (bootstrapped) 95% conﬁdence intervals that are diﬀerent from zero (see ﬁgure 2), indicating that only σ-Mv has a signiﬁcant eﬀect on the Adaptation Score. We therefore believe that our initial assumption, in which we stated that speciﬁcally movement speed is beneﬁcial for temporal adaptation, is not violated. However, we understand that other readers might have these same questions, which is why we have added the results of the above mentioned PLS to the manuscript and discussed its implications further in the discussion section.

2. In the adaptation, we can assume a simple state-space model,

Mv(t + 1) = λ ∗ Mv(t) + η ∗ (TE(t) + ζ(t)) + xi(t)

where lambda is forgetting rate, eta is learning rate, xi(t) is motor noise, and zeta(t) is sensory noise to perceive TE(t). Under this equation, it is self-evident that the larger σ Mv results in faster learning, because a high magnitude of xi(t) + eta ∗ zeta(t) (i.e., σ Mv ) is nearly equivalent to high learning rate. Please mention clearly that the current results are not mere results of high sensory noise.

ANSWER: One important way in which adaptation can be modeled is with a state space model. Another way it can be modelled is with a Bayesian tool, for example a Kalman ﬁlter. The reason that these Bayesian of models are used is that learning from errors is often thought to be optimal. More uncertain feedback (due to sensory and/or motor noise) leads to smaller error updates. Although higher sensory noise generally leads to more variability in movement outcome, we would not expect learning rate to be higher because of it. Furthermore, assuming that zeta would be gaussian noise with mean of zero, we are not sure if the error would always be perceived as larger and therefore updates would be larger. The measure of error (TE) in our task is 1 dimensional. This means that the noise could have decreased the perceived error just as likely as it could have increased it, which means on average increased correction of the error is not expected. We have extended the text about noise in the discussion a bit to also cover sensory noise.

3. I wonder how generalization aﬀected the results. Because the authors used three diﬀerent speeds for target motions, they need to coednsider how the adaptation in one speed can be generalized to other speeds. Another possibility was the participants with small σMv showed little generalization compared to those with large σMv .

ANSWER: The three diﬀerent speeds were used to prevent participants from learning were to intercept at what time, without taking the movement of the target into account. Given the size of the dataset, we cannot be sure if the diﬀerent speeds have an eﬀect on the results. Some participants might have larger generalization than others (with the term generalization referring to how the mapping between action and consequence generalizes to other tasks/speeds). It is possible that at the moment the perturbation was applied, the broadly generalizing participants were able to generalize the changing consequences better for all target speeds and adapt faster as a result. However, higher generalization would not directly lead to more variable movement speed during the baseline.

On the other hand, increased movement speed variability might in itself lead to more generalization (as proposed by the reviewer’s second suggestion). Hence, more variable people might develop broader tuning curves, and as a result adapt faster. We think this idea is captured well within the notion of exploration that is part of our original hypothesis. Research has shown that generalisation temporal perturbations is limited only to tasks that are very similar to the learned task (de la Malla et al, 2014). Higher variability due to exploration might broaden the brains knowledge on the temporal contingencies over a broader range of target speeds and situations. Unfortunately our current dataset cannot give a clear answer to the question of generalization. As discussed in manuscript, we suspect the variability is a result of error correction rather than exploration. However, we understand that others might have the same question and have therefore dedicated a section to it in the discussion.

4. Please discuss slope and asymptote separately. Why did the authors sum those values to quantify Adaptation Score? 

ANSWER: Adaptation is often calculated by ﬁtting an exponential function on the data and calculating the Time Constant (TC) of this best-ﬁt. A smaller TC indicates better adaptation. This method assumes that adaptation takes place according to an exponential function. We aimed at another way of calculating adaptation, as during pilot studies we were not able to calculate this ﬁt for each participant, indicating that not each participants adaptation can be quantiﬁed this way (See ﬁgure 3 PDF). We opted for an Adaptation Score (AS) that could be determined for each participant so that we could also account for the participants that had less typical types of adaptation, for example more linear. The reason we normalised and summed the two in order to calculate the AS is because both slope and asymptote determine the quality of the adaptation. Very low levels of adaptation (asymptote, b) might be achieved with a slope (a) that is steep, while a higher level of adaptation might be achieved with a low slope. It is the participants that have both fast (slope) and high (asymptote) adaptation that have high AS, while slow and low adaptation would lead to low AS. Although we are unsure if this is the best way of quantifying the adaptation, we have found that (for the participants we were able to calculate the TC of the exponential for; n=11), the TC was negatively correlated with the AS (albeit not signiﬁcantly; R = - 0.54; p = 0.088). This indicates that the AS can replace TC as a measure of adaptation. Figure 4 below shows how the AS and TC are related. When we use the TC instead of the AS to correlate the relationship between σMv and adaptation, we ﬁnd a signiﬁcantly negative correlation (R = -0.63, p = 0.038), providing more evidence that our measure of adaptation yields similar results to the more standard ones. We have added these results to the manuscript.

5. Despite the clarity of Fig. 3B (if the Adaptation Score were a valid value to quantify adaptation), Figs. 4-6 do not seem meaninful. The reason is the small correlation in Fig.3, cross-correlation that does not seem to be signiﬁcantly diﬀerent from 0 in Fig. 4, and low correlations in Fig.5. In Figs. 3 and 5, it seems impossible to predict one variable based on the other. 

ANSWER: We acknowledge that we have not reached a deﬁning answer to the question of what the reason behind the variabilityadaptation relationship is. The data hints towards increased variability being due to the use of movement speed in order to correct for error. We aimed to convey the idea that the results regarding the cause of the relationship are non-conclusive, however we might not have fully succeeded. We will state more clearly that further research is needed to ﬁnd more conclusive answers.

References:

de la Malla, C., Lopez-Moliner, J., & Brenner, E. (2014). Dealing with delays does not transfer across sensorimotor tasks. Journal of vision, 14(12), 8-8.

Reviewer 2:

The authors examine the relationship between variability in hand movement speed during baseline and the adaptation to a temporal perturbation of sensory feedback (delay). They ﬁnd that baseline variability is a good predictor of adaptation expressed as a summary score including the rate at asymptote level of adaptation. The authors argue that the increased baseline variability seen in good adapters is not due to exploration, but rather to increased correction of errors in previous trials.

The source and relevance of motor variability, and the explanation of individual diﬀerences in motor learning, are critical issues in the ﬁeld of motor control and learning. The authors provide important novel insights into the role of exploration and variability in motor learning. The experiment is well designed, and the paper is generally well written and concise.

ANSWER: Dear reviewer 2, thank you for your kind words and your suggestions. We found them very valuable when rewriting the manuscript. We hope to have answered your questions and concerns to your satisfaction.

1. My main comment is that the analyses of the relation between variability and adaptation should be more clearly explained and motivated. How/why were the particular analyses performed? Why was the adaptation score calculated this way? What is meant by sequential eﬀects? How were the participants divided into groups? Why was the linear model ﬁtted to the group data, and the cross-correlation performed per participant? Why are both a Pearson and a Spearman correlation reported; how exactly are variability and adaptation related? Why do some analyses test diﬀerences between groups, while other analyses focus on correlation of baseline temporal variability with other measures? 

ANSWER: We have expanded the methods section to more clearly to account for the uncertainties and questions of the reviewer. In summary, we have added the use of an exponential function, more widely used in adaptation, and found a correlation with our score (see Fig 3 and 4). Once we have shown this correlation, we keep our score because, unlike the exponential function, we were able to ﬁt our score to all subjects. Concerning the use of the ccf, it only makes sense to apply to individual data, since averaging would factor out the individual corrective patters. This analysis has been done in the spatial domain (see van Beers, 2009). While the lmm can capture individual variability in the random structure of the model. We used Spearman, in addition to Pearson, because Spearman (based on ranks) captures monotonic relation better than Pearson (which requires linear relations). We have rewritten the methods to motivate our choices more clearly.

2. Another important point is that the term exploration should be clearly deﬁned. Is this an actively controlled (or deliberate) process, or an automatic, implicit process? The term exploration strategies (used by the authors) makes me think that it is a deliberate process, but I am not sure this is what the authors mean. This part could indeed use some clariﬁcation. ANSWER: The process is thought to be actively controlled. We have rewritten the section (note that Q6 is incorporated in this section as well).

3. Lines 4-18. I understand what the authors are saying here but the reasoning is a little confusing. They ﬁrst argue that we need sensory feedback, and then say that sensory feedback is too slow for corrections and we thus need an internal model. It might be helpful to distinguish between immediate corrections and adjustment to changes. In addition, we also need to predict the trajectory of the ball if were intercepting a moving object and rely on sensory feedback. Please clarify this paragraph. ANSWER: Agreed: we have rewritten the paragraph.

4. L 22. neural variability. If the authors refer to execution noise as well as planning noise, please remove neural. ANSWER: Corrected

5. L 30-31. In situations in which error feedback is not clear I think not available would be the correct term here. If not, please explain what is meant by not clear. ANSWER: Corrected to ”unavailable or uninformative”

6. L 33-35. It can also be beneﬁcial . in the explored dimension of the behaviour. Its not clear to me what this means, please expand. ANSWER: Changed in order to increase clarity.

7. L 36-38. Since the Wu et al paper is a main motivation for the current study, I think it would be helpful to expand the description of their ﬁndings a little bit. ANSWER: Expanded

8. L 42-45. It is not clear to me how exploration strategies could directly beneﬁt the updating of the forward model, please clarify. And what is the potential role of variability from other sources than exploration? ANSWER: Exploration strategies can help us learn about the outcome of our motor commands. When we explore more, we get richer information about the relationship between the motor commands and its sensory outcome. We have expanded our explanation of exploration. We also added some more information about other forms of noise, like sensory noise.

9. Fig 1B. Display 4 is missing a title. It is unclear what the ﬁgure below the 4 displays belongs to. ANSWER: Corrected

10. L 92-93. What was the size of the target and the cursor? ANSWER: Added

11. L 109-110. 0.5 cm from where? ANSWER: Added

12. Fig 2 - legend. The word summarized should be summed. ANSWER: Corrected

13. Were the results collapsed for the diﬀerent target speeds? ANSWER: We indeed did not look at the diﬀerences in target speed. Target speed was one of 3 speeds leftwards or rightwards random. We chose diﬀerent target speeds so that the participants could not just learn a speciﬁc location and time to intercept at each trial. However, our dataset is not large enough to do the analysis for diﬀerent. However, in accordance with the comments of reviewer 1 (Q3), we elaborated about the use of the diﬀerent speeds and its aﬀect on generalization in the discussion.

14. L 177. There seems to be a slightly faster adaptation for the HIGH group than for the LOW group. This is not clear to me from ﬁgure 3A. Was the slope or asymptote level of adaptation diﬀerent between groups? ANSWER: This indeed was initially only visual inspection and was after veriﬁed with the Adaptation score. We rewrote this section.

15. Fig 4. I would just like to note that I dont ﬁnd this ﬁgure particularly convincing. From the scatter, it looks like there is no relationship in either of the groups. ANSWER: We acknowledge that we have not reached a deﬁning answer to the question of what the reason behind the variabilityadaptation relationship is. The data hints towards increased variability being due to the use of movement speed in order to correct for error. We aimed to convey the idea that the results regarding the cause of the relationship are exploratory, however we might not have fully succeeded. We will state more clearly that the data of the second part is exploratory and should be aimed to be reproduced in future studies.

16. L 206-207. Shouldnt the lag zero correlation be positive for both groups? ANSWER: Lag zero cross correlation was more positive in the HIGH variability group. The HIGH variability group moved in more variable speeds and it might be that the relationship between movement speed and error might therefore more easily revealed. It could also mean that the HIGH variability group had a stronger relationship between movement speed and error, they were more likely to use movement speed as a correction mechanism in future. We expanded this part in the discussion a bit further.

17. Fig 5. What do the error bars represent? ANSWER: Error bars represent the 95% bootstrapped conﬁdence intervals. We have now added this in the ﬁgure description as well.

18. L 213-214. A word is missing in this sentence. ANSWER: Corrected

19. Could the authors speculate about the origin of diﬀerences in error correction? For example, do people have diﬀerences in error sensitivity? Is the mechanism similar to diﬀerences in adaptation rate resulting from diﬀerences in error size or diﬀerent levels of uncertainty of sensory feedback? 

ANSWER: One of the mentioned mechanisms in the Discussion is a possible diﬀerence in planning noise, which might lead to more uncertain predicted sensory feedback (hence more correction of error) or enhance the detection of errors. We expanded a bit on that topic and added the role of sensory noise. We also included a part about a possible role of generalization on error correction and the idea that error can be interpreted as having a more spatial or more temporal origin.

---

## [Decision Letter · Decision Letter 1]

19 Nov 2019

PONE-D-19-18039R1

Increased error-correction leads to both higher levels of variability and adaptation

PLOS ONE

Dear Dr. López-Moliner,

Thank you for submitting your manuscript to PLOS ONE. After careful consideration, we feel that it has merit but does not fully meet PLOS ONE’s publication criteria as it currently stands. Therefore, we invite you to submit a revised version of the manuscript that addresses the points raised during the review process.

The two expert reviewers agree that the paper has merit but had some final comments that I would like to see addressed as much as possible in a revised manuscript. The comments/requests are relatively minor and I believe that I will be able to reach a final decision without additional input from the reviewers.

We would appreciate receiving your revised manuscript by Jan 03 2020 11:59PM. To enhance the reproducibility of your results, we recommend that if applicable you deposit your laboratory protocols in protocols.io, where a protocol can be assigned its own identifier (DOI) such that it can be cited independently in the future. For instructions see: http://journals.plos.org/plosone/s/submission-guidelines#loc-laboratory-protocols

We look forward to receiving your revised manuscript.

Kind regards,

Welber Marinovic

Academic Editor

PLOS ONE

Reviewers' comments:

Reviewer's Responses to Questions

**Comments to the Author**

1. If the authors have adequately addressed your comments raised in a previous round of review and you feel that this manuscript is now acceptable for publication, you may indicate that here to bypass the “Comments to the Author” section, enter your conflict of interest statement in the “Confidential to Editor” section, and submit your "Accept" recommendation.

Reviewer #1: All comments have been addressed

Reviewer #2: (No Response)

2. Is the manuscript technically sound, and do the data support the conclusions?

Reviewer #1: Yes

Reviewer #2: Yes

3. Has the statistical analysis been performed appropriately and rigorously? 

Reviewer #1: Yes

Reviewer #2: Yes

4. Have the authors made all data underlying the findings in their manuscript fully available?

Reviewer #1: Yes

Reviewer #2: Yes

5. Is the manuscript presented in an intelligible fashion and written in standard English?

Reviewer #1: Yes

Reviewer #2: Yes

6. Review Comments to the Author

Reviewer #1: The authors revised their manuscript carefully. Although they missed discussing the effects of generalization, their citing paper can support to interpret the effects of generalization to some extent (but not wholly, of course). I have now a positive attitude towards the acceptance of this manuscript after responding to the following comments. I have two more comments about the PLS, the evaluation of the adaptation score, and discussion.

1) Although I understand that they want to focus on the adaptation score, the temporal error (TE) is a more direct measure of adaptation without any post-hoc calculation of such as variability. Although I understand the difficulty of calculating the relation among TE, movement angle (ma), and movement velocity (mv), recent studies enable us to evaluate the relation using a data-driven technique [1]. Further, it is possible to calculate the task-relevant and task-irrelevant variabilities with the technique. It should be better to discuss the relation between task-relevant variability and adaptation score after clarifying the relation among TE, ma, and mv.

In the analysis of PLS, they considered only a simple second-order interaction of ma and mv. I guess that the TE can be affected by mv*T*sin(ma) under the assumption of the constant velocity through the movement time T (although this assumption is probably wrong). It should be investigated the more appropriate relation between adaptation score and some kinematic variables.

2) In l.72-76, they mentioned that the history of error of future error could affect adaptation. Why not consider the influence of predicted error on motor adaptation [2]? Because we cannot be sure what kind of information affects motor adaptation, we should keep our scope broad.

ref:

[1] Furuki D & Takiyama K, 2019, Decomposing motion that changes over time into task-relevant and task-irrelevant components in a data-driven manner: application to motor adaptation in whole-body

movements, Sci Rep

[2] Takiyama K, Hirashima M, Nozaki D, 2015, Prospective errors determine motor learning, Nat Comm

Reviewer #2: I am happy with most of the changes that the authors made. However, I do have a few more comments, as outlined below. Line numbers refer to the manuscript with tracked changes.

Line 195-197. “The slope (a) and asymptote (b) that resulted in a minimum residuals were first normalized across subjects and then summed.” I realized that the question that I asked in the previous round, “Why was the adaptation score calculated this way?”, was rather vague. I am happy that the authors clarified why they used this score rather than the time constant of an exponential function. However, I also meant to ask whether the authors expected variability to influence both the slope and the asymptote of adaptation? What was the reason for summing the slope and asymptote, rather than treating them as individual variables?

Line 214-223. After reading the comment of the other reviewer on the redundancy of the task, I understand why the authors performed this analysis, but this analysis is not clearly motivated in the paper. (I also think that in its current form this analysis will not mitigate the concern of the other reviewer, but I will leave this up to the other reviewer. I’d be happy to explain more if needed).

This analysis makes me wonder about the strategies that participants used. Were most participants fairly stereotyped in where they would intercept the target? The figure showing the Mv and Ma variability correlations seems to suggest that most participants had a fairly low variance in Ma, but some participants have a rather high variance.

It would also be useful to explain the relationship between Mv and Ma in more detail than mentioning that there is a correlation. Have the authors verified whether this relation is independent of target speed?

Line 257. “Adaptation scores varied between -1.49 and +1.67.” Since the scores are normalized these numbers don’t provide much information. It would be useful to have an overview of the (unnormalized) slopes, asymptotes and adaptation scores for all participants. It would also be useful to provide some information about the goodness of fit for the two-state lines.

Figure 2 + 4. The y-axis of these figures is labelled ‘Temporal Error (ms)’. This might be confusing as it suggests that the error increases during the adaptation phase (rather than going towards zero). In addition, it seems that the numbers on the y-axis of the revised Fig 2 are seconds instead of ms.

7. PLOS authors have the option to publish the peer review history of their article (what does this mean?). If published, this will include your full peer review and any attached files.

Reviewer #1: No

Reviewer #2: No

---

## [Author Response · Author response to Decision Letter 1]

2 Jan 2020

Review Comments to the Author  Please use the space provided to explain your answers to the questions above. You may also include additional comments for the author, including concerns about dual publication, research ethics, or publication ethics. (Please upload your review as an attachment if it exceeds 20,000 characters)

  Reviewer #1: The authors revised their manuscript carefully. Although they missed discussing the effects of generalization, their citing paper can support to interpret the effects of generalization to some extent (but not wholly, of course). I have now a positive attitude towards the acceptance of this manuscript after responding to the following comments. I have two more comments about the PLS, the evaluation of the adaptation score, and discussion. 

Dear reviewer, thank you for your comments. Below you can find the answers to other points raised. We hope we have sufficiently satisfied your concerns.

 1) Although I understand that they want to focus on the adaptation score, the temporal error (TE) is a more direct measure of adaptation without any post-hoc calculation of such as variability. Although I understand the difficulty of calculating the relation among TE, movement angle (ma), and movement velocity (mv), recent studies enable us to evaluate the relation using a data-driven technique [1]. Further, it is possible to calculate the task-relevant and task-irrelevant variabilities with the technique. It should be better to discuss the relation between task-relevant variability and adaptation score after clarifying the relation among TE, ma, and mv.In the analysis of PLS, they considered only a simple second-order interaction of ma and mv. I guess that the TE can be affected by mv*T*sin(ma) under the assumption of the constant velocity through the movement time T (although this assumption is probably wrong). It should be investigated the more appropriate relation between adaptation score and some kinematic variables.

We agree with the suggestion to check the relationship between different kinematic variables and TE. We used a LASSO regression to analyse explanatory predictors (similar to the Ridge regression in the suggested literature). This showed us that the movement time was one of the main predictors of Temporal error. We already saw that To not exhaust the reader with too many different types of analyses we have also changed the PLS to a LASSO regression to see the relationship between sigma_mv, sigma_ma and adaptation.  2) In l.72-76, they mentioned that the history of error of future error could affect adaptation. Why not consider the influence of predicted error on motor adaptation [2]? Because we cannot be sure what kind of information affects motor adaptation, we should keep our scope broad.

 We agree with the reviewers comment and have added some information about prospective errors in the introduction.

ref: [1] Furuki D & Takiyama K, 2019, Decomposing motion that changes over time into task-relevant and task-irrelevant components in a data-driven manner: application to motor adaptation in whole-body movements, Sci Rep [2] Takiyama K, Hirashima M, Nozaki D, 2015, Prospective errors determine motor learning, Nat Comm  Reviewer #2: I am happy with most of the changes that the authors made. However, I do have a few more comments, as outlined below. Line numbers refer to the manuscript with tracked changes. Dear reviewer, thank you very much for your further comments. We hope we have satisfied your final notes.

  Line 195-197. “The slope (a) and asymptote (b) that resulted in a minimum residuals were first normalized across subjects and then summed.” I realized that the question that I asked in the previous round, “Why was the adaptation score calculated this way?”, was rather vague. I am happy that the authors clarified why they used this score rather than the time constant of an exponential function. However, I also meant to ask whether the authors expected variability to influence both the slope and the asymptote of adaptation? What was the reason for summing the slope and asymptote, rather than treating them as individual variables? Although we were interested to see if there was a difference, we did not have a specific prediction. Rather we tried to find a replacement for the more commonly used exponential function. We chose to sum the two because summing them leads to an Adaptation Score that represents fast and high levels of adaptation with high scores, and slow and low levels of adaptation with low scores. Anything in between can either mean an interplay between the slope and intercept. In some previous literature the focus was put on speed of adaptation to benefit from exploration strategies. Looking at the individual slopes and asymptotes we unfortunately cannot make up any distinct effect. This might also be because we had only a few participants with higher slopes. We have added an overview of the slopes and asymptotes.

Line 214-223. After reading the comment of the other reviewer on the redundancy of the task, I understand why the authors performed this analysis, but this analysis is not clearly motivated in the paper. (I also think that in its current form this analysis will not mitigate the concern of the other reviewer, but I will leave this up to the other reviewer. I’d be happy to explain more if needed). We have expanded our motivation on the reasons for the performed analyses.

 This analysis makes me wonder about the strategies that participants used. Were most participants fairly stereotyped in where they would intercept the target? The figure showing the Mv and Ma variability correlations seems to suggest that most participants had a fairly low variance in Ma, but some participants have a rather high variance.It would also be useful to explain the relationship between Mv and Ma in more detail than mentioning that there is a correlation. Have the authors verified whether this relation is independent of target speed? We have added a Lasso regression analysis to look at how TE was affected by different kinematic variables, among others: target speed, Mv and Ma.

Line 257. “Adaptation scores varied between -1.49 and +1.67.” Since the scores are normalized these numbers don’t provide much information. It would be useful to have an overview of the (unnormalized) slopes, asymptotes and adaptation scores for all participants. It would also be useful to provide some information about the goodness of fit for the two-state lines.

 See above

Figure 2 + 4. The y-axis of these figures is labelled ‘Temporal Error (ms)’. This might be confusing as it suggests that the error increases during the adaptation phase (rather than going towards zero). In addition, it seems that the numbers on the y-axis of the revised Fig 2 are seconds instead of ms. Changed Temporal Error to Temporal Hand Error in order to clarify its meaning. Changed all TE measurements to ms.

---

## [Editor Report · Decision Letter 2]

3 Jan 2020

Increased error-correction leads to both higher levels of variability and adaptation

PONE-D-19-18039R2

Dear Dr. López-Moliner,

We are pleased to inform you that your manuscript has been judged scientifically suitable for publication and will be formally accepted for publication once it complies with all outstanding technical requirements.

With kind regards,

Welber Marinovic

Academic Editor

PLOS ONE
---

## [Editor Report · Acceptance letter]

8 Jan 2020

PONE-D-19-18039R2 

Increased error-correction leads to both higher levels of 2 variability and adaptation 

Dear Dr. López-Moliner:

I am pleased to inform you that your manuscript has been deemed suitable for publication in PLOS ONE. Congratulations! Your manuscript is now with our production department. 

With kind regards,

on behalf of

Dr. Welber Marinovic 

Academic Editor

PLOS ONE